# Exploring Endophytic Bacteria from *Artemisia* spp. and Beneficial Traits on Pea Plants

**DOI:** 10.3390/plants13121684

**Published:** 2024-06-18

**Authors:** Shervin Hadian, Donald L. Smith, Stanislav Kopriva, Eglė Norkevičienė, Skaidrė Supronienė

**Affiliations:** 1Microbiology Laboratory, Lithuanian Research Centre for Agriculture and Forestry, Institute of Agriculture, Instituto Ave. 1, Akademija, LT-58344 Kėdainiai, Lithuania; skaidre.suproniene@lammc.lt; 2Department of Plant Science, McGill University, Montreal, QC H9X 3V9, Canada; donald.smith@mcgill.ca; 3Botanical Institute, University of Cologne, 50674 Cologne, Germany; skopriva@uni-koeln.de; 4Department of Grass Breeding, Lithuanian Research Centre for Agriculture and Forestry, Institute of Agriculture, Instituto Ave. 1, Akademija, LT-58344 Kėdainiai, Lithuania; egle.norkeviciene@lammc.lt

**Keywords:** *Artemisia* sp., *Fusarium* sp., nitrogen fixation, pea, phosphate solubilization

## Abstract

Endophytic microorganisms represent promising solutions to environmental challenges inherent in conventional agricultural practices. This study concentrates on the identification of endophytic bacteria isolated from the root, stem, and leaf tissues of four *Artemisia* plant species. Sixty-one strains were isolated and sequenced by 16S rDNA. Sequencing revealed diverse genera among the isolated bacteria from different *Artemisia* species, including *Bacillus*, *Pseudomonas*, *Enterobacter*, and *Lysinibacillus*. AR11 and VR24 obtained from the roots of *A. absinthium* and *A. vulgaris* demonstrated significant inhibition on *Fusarium* c.f. *oxysporum* mycelial growth. In addition, AR11, AR32, and CR25 exhibited significant activity in phosphatase solubilization, nitrogen fixation, and indole production, highlighting their potential to facilitate plant growth. A comparative analysis of *Artemisia* species showed that root isolates from *A. absinthium*, *A. campestris*, and *A. vulgaris* have beneficial properties for inhibiting pathogen growth and enhancing plant growth. AR11 with 100% similarity to *Bacillus thuringiensis*, could be considered a promising candidate for further investigation as microbial biofertilizers. This finding highlights their potential as environmentally friendly alternatives to chemical pesticides, thereby contributing to sustainable crop protection practices.

## 1. Introduction

Modern agriculture encounters numerous challenges, including biotic stresses from pathogens and pests, and abiotic factors such as salinity, drought, and extreme temperatures. These issues are exacerbated by the continuous rise in the global population. The combined effects of extreme climatic events, like droughts, floods, and high temperatures, along with the unsustainable use of water resources, have led to substantial reductions in crop yields. Furthermore, agriculture also faces problems related to the loss of biodiversity and the excessive use of chemical pesticides and fertilizers. The overuse of chemical compounds resulted in pathogen resistance and a polluted environment [1]. Recent research efforts focus on developing alternative methods to reduce reliance on chemical compounds.

Endophytic bacteria colonize and inhabit the internal tissues of plants. The positive impacts of endophytes on improving plant growth and health have been extensively studied [2]. Endophytic bacteria contribute to enhanced plant growth through processes such as nitrogen fixation [3], stress alleviation [4], production of phytohormones [5], and the generation of growth-promoting signal compounds [6]. Endophytic bacteria have the capability to induce tolerance to pollutants, making them potential candidates for application in bioremediation [7]. Beyond terrestrial plants, numerous studies have pointed out that culturable endobacteria in wetland plants can serve various purposes, including bioremediation, enzyme production [8], biocontrol [9], and promotion of plant growth [10].

*Artemisia* L. plants are known for their many beneficial bioactive compounds, which have antimicrobial, antifungal, antioxidant, and allelopathic effects. Furthermore, endophytic bacteria associated with *Artemisia* plants have been identified as contributors to various advantages for their host plants. These benefits encompass improved nutrient absorption, regulation of plant phytohormones, and the ability to combat pests and pathogens. Despite extensive research on the bioactive compounds of *Artemisia* plant extracts, there is a significant gap in understanding the diversity of endophytic bacteria associated with *Artemisia* spp. and their mechanism of action [11].

The two endophytic root bacteria, *Bacillus subtilis* and *Stenotrophomonas* spp. are associated with *Artemisia annua*. They enhance the host plant’s nitrogen and phosphorus absorption, resulting in increased shoot height, leaf area, and overall plant biomass [12]. Ebu et al. (2023) reported the isolation of various endophytic bacteria, including *Bacillus cereus*, *B. subtilis*, *Citrobacter freundii*, *Enterobacter asburiae*, *E. cloacae*, *E. kobei*, *E. ludwigii*, *Enterococcus faecium*, and *Pseudomonas monteilli* from *Artemisia annua*, *Moringa oleifera*, and *Ocimum lamiifolium*. They found that these endophytic bacteria have the potential to solubilize insoluble forms of phosphate and zinc [13].

*Fusarium oxysporum* is a soil-borne phytopathogen that causes vascular wilt, root rot, and crown rot diseases on a wide range of economically significant crop plants, including tomato, banana, sweet potato, cotton, pea, and chickpea [14,15]. One of the most serious pea diseases is root rot, which is caused by *F. oxysporum* [16]. It caused severe damage at all stages of crop growth, with yield losses of up to 97% [17]. Endophytic bacteria isolated from various higher plants, including tomato, guava, bamboo, pine, and cacao, demonstrated significant antifungal activity against *Fusarium oxysporum* in tomato. Additionally, these bacteria were found to directly enhance the growth of tomato plants [18,19].

The present study aimed to determine the diversity of endophytic bacteria living in different *Artemisia* spp. plant tissues (*A*. *absinthium*, *A. campestris*, *A. dubia*, and *A. vulgaris*) by 16S rDNA gene sequence, followed by phylogenetic analysis. Evaluate their potential to inhibit the mycelial growth of pea root rot causing *Fusarium* sp. Assess their beneficial traits such as phosphate solubilization, nitrogen fixation, indole production, and seed germination enhancement. We emphasized the role of endophytic bacteria associated with various *Artemisia* species in the development of biocontrol agents aimed at reducing reliance on chemical fungicides and promoting eco-friendly agricultural practices.

## 2. Results

### 2.1. Isolation Endophytic Bacteria from Artemisia spp.

In total, 61 endophytic bacteria displaying distinct morphologies were isolated from the roots, stems, and leaves of *A. absinthium*, *A. campestris*, *A. dubia*, and *A. vulgaris* plants. Morphological and biochemical characterization of the individual strains showed that bacteria species isolated from different plant parts of *Artemisia* species were distinct, though some similarities were noted. Morphologically, the endophytic bacteria isolates displayed a variety of colony shapes, colors, and margins (Figure 1). Gram staining revealed that 83% of isolates were Gram-positive. Sixty-one percent of the isolates showed positive catalase activity.

### 2.2. Molecular Identification of Endophytic Bacteria

Based on 16S rDNA gene sequencing, the majority of strains belonged to the phylum *Proteobacteria* and *Firmicutes*. These sequences were subsequently used for phylogenetic analyses, as shown in Figure 2. Specifically, strains isolated from *A. absinthium* (AR11, AR32, and AR35) showed similarity to *Bacillus thuringiensis* IAM12077, *B. cereus* ATCC14579, and *B. velezensis* FZB42. Strains from *A. campestris* (CR25 and CL14) exhibited similarity to *Pseudomonas fluorescens* IAM 12022 and *B. thuringiensis* IAM12077. Strains from *A. dubia* (DR31 and DR33) displayed similarity to *B. amyloliquefaciens* NBRC15535 and *B. subtilis* IAM12118. Strains from *A. vulgaris* (VR32 and VR24) showed similarity to *B. velezensis* FZB42 and *B. thuringiensis* IAM12077.

### 2.3. Inhibitory Effect of Endophytic Bacteria on the Growth of Fusarium c.f. oxysporum

Phytopathogenic fungi were isolated from pea roots exhibiting disease symptoms. The ability of the isolates to cause root rot in pea plants was confirmed by plant pathogenicity assays. Morphological analysis and ITS sequence revealed high sequence similarity to *Fusarium oxysporum* HDV247. The in vitro growth inhibition activity test *F.* c.f. *oxysporum* showed that 61 endophytic bacteria differed in their inhibitory effects on the growth of *F.* c.f. *oxysporum* mycelia (Figure 3).

Statistical analysis showed significant differences (*p* < 0.001) between the growth inhibition activity of 61 endophytic bacteria on *F.* c.f. *oxysporum* AR11 (90.83%) and VR24 (88%) strains isolated from the roots of *A. absinthium* and *A. vulgaris*, respectively, showed the highest inhibitory effect on *F.* c.f. *oxysporum* growth (Figure 4). 

The mean growth inhibition activity values across different plant species showed that strains from *A. absinthium* exhibited the highest levels, followed by strains from *A. dubia*, *A. vulgaris*, and *A. campestris*, respectively (Figure 5). *A. campestris* displayed a wider range of values and growth inhibition activity compared to other *Artemisia* species. *A. vulgaris* displayed a more limited range of values for growth inhibition.

### 2.4. Phosphate Solubilization Index

The phosphate solubilization was assessed on Pikovskaya agar. Several isolates exhibited distinct halo zones around their colony (Figure 6). The halo diameter varied among the different strains.

The phosphate solubilization index (PSI) values represent the ability of these endophytic bacteria to solubilize phosphate over time, showing that 36% of isolates demonstrated phosphate solubilization activity. Specifically, 61% of the isolates from *A. absinthium*, 35% from *A. campestris*, 27% from *A. dubia*, and 22% from *A. vulgaris* showed phosphate solubilization activity (Figure 7).

Strain AR11 displayed the highest PSI after 10 days of incubation (2.93) whereas isolates from *A. campestris* showed lower PSI values compared to those from *A. absinthium*.

### 2.5. Biochemical Properties and Effect of the Endophytic Bacteria on Pea Plant Growth

In this study, 20% of the isolated bacteria exhibited positive results in the indole production test, while 44% demonstrated nitrogen fixation potential. Based on these and other test results, eight isolated strains were selected to study their effects on pea seed germination and growth promotion. The selected isolates are listed in Table 1.

The analysis of seed germination across various endophytic bacteria revealed that germination started within 48 h. The inoculation with VS32 suppressed pea seed germination. However, the seeds inoculated with AR11 and AR32 exhibited 100% seed germination rates. VL13 showed the least effect on seed germination, with only a 78% rate observed. There were no significant differences between isolates for seed germination.

Although some strains, like CR25, AR11, AR32, and VR24 did not exhibit a significant increase in root length, there was significant lateral root branching. DR31 not only did not make any significant effects on increasing shoot and root length but also demonstrated no positive impact on root branching (Figure 8).

AR11 and CR25 exhibited the most significant (*p* < 0.05) impact on root and shoot length compared to the control. While AR32 had a smaller effect on root length than CR25 (no significant difference), it displayed a greater effect on shoot length compared to CR25. DR31 did not exhibit any significant effect on pea root and shoot length compared to the control (Figure 9).

## 3. Discussion

Endophytic bacteria play a crucial role in influencing the essential functions of host plants. They can enhance plant growth and induce resistance against pathogens. Utilizing endophytes in crop production has the potential to decrease the need for fertilizers and pesticides, thereby enhancing the sustainability of crop cultivation [20]. Non-native endophytes have been studied for their potential use as biofertilizers. For example, researchers have evaluated endophytic rhizobia from clover roots for their effectiveness in enhancing rice growth as biofertilizers [21]. In this research, we studied how non-native endophytic bacteria isolated from *Artemisia* sp. suppress *Fusarium* c.f. *oxysporum* extracted from pea root and promote pea plant growth.

### 3.1. Identification of Isolated Endophytic Bacteria

Plants greatly benefit from endophytic bacteria, which stay in host plant tissues and have no negative effects [22]. Medicinal plants like *Artemisia annua* can host a variety of functionally significant endophytic bacteria in their leaves, stems, and roots [23]. Ashitha et al. (2019) reported a diverse community of endophytic bacteria, including *Arthrobacter* sp., *Pseudomonas* sp., *Microbacterium* sp., *Psychrobacter* sp., *Enterobacter* sp., *Bacillus* sp., *Kosakonia cowanii*, and *Bacillus* sp. associated with the medicinal plant *A. nilagirica* [24]. These findings demonstrate the remarkable diversity of endophytic bacteria found in various *Artemisia* species. In our investigation, we observed that the majority of effective endophytic bacteria belonged to the *Bacillus* and *Pseudomonas* genera. Among the endophytic bacteria identified, *Bacillus cereus*, *B. subtilis*, and *B. thuringiensis* were commonly found across all four *Artemisia* species, indicating their widespread presence in these plant ecosystems. *B. altitudinis* and *B. mycoides* were mostly associated with *A. absinthium.*

### 3.2. Antifungal Activity of Endophytic Bacteria

In our study, isolates that exhibited significant growth inhibition of *Fusarium* c.f. *oxysporum* were identified as *Bacillus thuringiensis*, *B. cereus*, *B. velezensis*, *B. amyloliquefaciens*, and *Pseudomonas fluorescens*. The antagonistic activity of these endophytic bacteria against *F. oxysporum* has been well-documented. For instance, *B. thuringiensis* has been reported to inhibit the growth of *F. oxysporum* in peanut and onion plants [25,26], *B. cereus* strains in tomato [27], *B. velezensis* in cucumber [28], *B. amyloliquefaciens* in banana [29], and *Pseudomonas fluorescens* in watermelon [30]. These findings align with our results, highlighting the potential of these endophytic bacteria as effective biocontrol agents against root rot pathogens.

### 3.3. Plant Growth-Promoting Characteristics

The biochemical properties of the isolated endophytic bacteria provided essential insights into their potential applications as plant growth-promoting agents. The capacity for nitrogen fixation and phosphate solubilization, along with indole production, were critical factors evaluated in this study. Strain AR11 exhibited strong phosphate solubilization, nitrogen fixation abilities, and increased plant growth. These multifunctional traits suggest that AR11 could be a plant growth promoter, aligning with previous studies that have highlighted *B. thuringiensis* as a powerful phosphate-solubilizing bacterium [31], nitrogen fixer [32] and effective plant growth-promoter [33]. Seed germination started within 48 h post-inoculation with selected endophytic bacteria, indicating their potential as biofertilizers. This is consistent with findings from previous studies where endophytic bacteria like *Bacillus* spp. and *Pseudomonas* spp. significantly enhanced seed germination and early seedling growth of tomatoes [34]. Conversely, strain VS32 suppressed pea seed germination, which suggests that not all endophytic bacteria have beneficial effects on germination and could have strain-specific or plant-specific interactions [35,36]. In terms of root development, strains CR25, AR11, AR32, and VR24 notably increased lateral root branching, although they did not significantly enhance root length. Lateral root branching is essential for establishing a resistant root system capable of better nutrient absorption [37]. The significant increase in root branching observed with CR25 (*Pseudomonas syringae*), indicates its substantial impact on root system architecture. This aligns with reported studies showing that *Pseudomonas* spp. modify root architecture and promote growth in *Arabidopsis thaliana* [38].

## 4. Materials and Methods

### 4.1. Plant Sample Collection

This study examined four species within the *Artemisia* genus: *A. absinthium* L., *A. campestris* L., and *A. dubia* Wall. ex-Besser, and *A. vulgaris* L. Among these, *A. absinthium*, *A. campestris*, and *A. vulgaris* are indigenous to the Lithuanian flora while *A. dubia* is introduced and cultivated. Plants from all tested species were collected in dry perennial anthropogenic herbaceous vegetation habitats at three distinct locations in Lithuania (Kaunas, Kėdainiai, Šiauliai) Figure 10, during the month of June 2021. Collected plants were at the growth stage of leaf development (main shoot). Subsequently, in the laboratory, the plant samples underwent thorough washing with water to eliminate all soil debris before being dried at room temperature. Surface disinfection was accomplished using aseptic immersion in 70% ethanol for 2 min after 5% sodium chloride. Subsequently, the samples were immersed in hypochlorite for 4 min and thoroughly rinsed using sterile distilled water, followed by a 2 min contact with 70% ethanol. Finally, after rinsing with sterile distilled water again, allowed to dry under sterile conditions. To ensure that all epiphytes were eliminated throughout the surface sterilization procedure, 100 µL of water from the final wash step was added onto the nutrient agar and incubated at 28 °C for 7 days and monitored for bacteria growth [17].

### 4.2. Isolation Endophytic Bacteria

Plant parts (stem, root, leaf) of each species that had been surface-sterilized were cut into small pieces. Endophytic microorganisms were isolated through a direct planting technique. Each plant part was cut into 1 × 1 cm sections and placed in Petri dishes containing TSA (tryptophan soy agar) medium. The plates were incubated at 28 °C for up to 15 days. On days 2, 5, 10, and 15, colonies were chosen and purified in TSA. Each plate was incubated for 14 days at a temperature of 27–29 °C. The colonies were chosen for each Petri dish based on their growth stage and morphology (color, size, and shape).

### 4.3. Molecular Identification of Isolated Endophytic Bacteria by 16srDNA

#### 4.3.1. DNA Extraction from Bacteria Strains

The DNA extraction from isolated bacteria followed the CTAB DNA extraction protocol provided by Prof. Thorsten Mascher’s lab [39]. Bacteria were cultured in a 10 mL LB medium and incubated at 37 °C with shaker speed at 200 rpm until reaching an optical density (OD600) of 0.8–1.0. The cells were then harvested by centrifugation at 5000 rpm for 10 min at room temperature (RT). The cell pellet was resuspended in 400 µL of TEN buffers (10 mM Tris/HCl pH 8.0, 10 mM EDTA, 150 mM NaCl) in a 2 mL Eppendorf tube, to which 20 µL of lysozyme was added and incubated at 37 °C for 20 min. Subsequently, 2 µL of RNase was added and incubated at 65 °C for 3 min. Afterward, 40 µL of SDS, 1 µL of proteinase K, and 550 µL of TEN* buffer (10 mM Tris/HCl pH 8.0, 1 mM EDTA, 50 mM NaCl) were added, mixed thoroughly, and incubated at 60 °C for 2 h. Next, 900 µL of phenol (equilibrated with TE buffer, pH 7.5–8.0) was added, and the mixture was inverted to ensure thorough mixing before centrifugation at 13,000 rpm for 5 min. The upper phase was carefully transferred into a new 1.5 mL Eppendorf tube. This extraction process was repeated once with phenol and twice with chloroform: isoamyl alcohol (24:1). The final aqueous phase was transferred to a test tube containing 10 mL of −20 °C cold ethanol, where the DNA precipitated. After air drying, the DNA was dissolved in 100 µL of TEN* buffer and incubated overnight at 4 °C. The DNA quality and quantity were assessed with Nanodrop and Qubit analyses.

#### 4.3.2. Taxonomic Identification

The bacteria 16S rDNA was amplified by PCR by universal primer pair 27F 5′-(AGAGTTTGATCMTGGCTCAG)-3′, and 1387R 5′-(GGGCGGWGTGTACAAG GC)-3′ [40]. The PCR amplification was conducted in a PCR thermal cycler (Bio-Rad My cycler, Berkeley, CA, USA), and comprised an initial denaturation at 95 °C for 5 min, followed by 25 cycles of 95 °C for 30 s, 56 °C for 30 s, 72 °C for 1.5 min, on the last cycle. The PCR amplicons were purified with the Gene JET PCR Purification Kit (Thermo Scientific, Dreieich, Germany) and the amplicons sequencing was conducted with Applied Biosystems 3730XL DNA Analyzer by BMR GENOMIC company (Padova, Italy). Retrieved DNA sequences were manually edited using Bio edit Sequence Alignment Editor Software (Version 7.2.5). The similarities of 16S rDNA genes between isolated endophytes and their phylogenetic neighbors were determined by using Basic Local Alignment Search Tool (BLAST) [41]. Multiple alignments were conducted using the CLUSTAL W Program in Bio edit Sequence Alignment Editor Software (Version 7.2.5) [42]. A phylogenetic tree based on 16S rRNA gene sequences was constructed by the neighbor-joining method, using a distance algorithm with a bootstrap of 1000, with Molecular Evolutionary Genetics Analysis (MEGA X) software version 11 [43]. The evolutionary distances were computed using the maximum composite likelihood method and were in the units of the number of base substitutions per site.

### 4.4. Pathogenic Fungi Isolation

Pathogenic fungi were obtained from diseased pea roots. Following surface sterilization, the samples were air-dried on sterile filter paper. Sections of roots, approximately 4 × 1 cm in size, were then placed onto Petri dishes containing potato dextrose agar (PDA) isolation medium. After obtaining pure cultures, they were placed at 25 °C for 5 days to grow before morphological evaluation. Microscopic observations of the two isolated fungal cultures were conducted using a light microscope and consisted of an evaluation of the septum and conidia size, shape, and color. Colony morphology was evaluated for colony color and growth pattern. After identification based on morphological characteristics, the ITS3f (5′-GCATCGATGAAGAACGCAGC-3′) and ITS4r (5′-TCCTCCGCTTATTGATATGC-3′) [44] primers were employed for molecular identification of the ITS region of pathogenic fungi. To assess the pathogenicity of each isolate, we utilized the pathogenicity test on pea plants [45].

### 4.5. Growth Inhibition Effect of Endophytic Bacteria on Pea Root Rot Pathogen

The growth inhibition activity of the isolated bacteria was evaluated by using the dual plate culture technique. Bacteria grown on the TSA for 24 h were used. The fungi were initially inoculated in the center of the plate with PDA medium and allowed to grow for one day. The bacteria strains were co-inoculated on the same agar plate at a distance of 2.5 cm from the fungal strains. The plates were then incubated at 28 °C for 7 days. The percentage growth inhibition of fungal mycelium compared to the control. This calculation was conducted after 7 days using the formula: (R1 − R2)/R1 × 100, where R1 and R2 represented the radial growth of the pathogen in the absence and presence of the antagonist, respectively.

### 4.6. Evaluation of Isolates for Their Plant Growth Promotion Potential

#### 4.6.1. Indole Production Test

The strains were inoculated to evaluate tubes with one 1% tryptone broth and were incubated at 28 °C for 48 h. Further, 1 mL of Kovac’s reagent (hydrochloric acid and *p*-dimethylaminobenzaldehyde in amyl alcohol) was added to each tube, the tubes were shacked continuously for 10 to 15 min, then, they were left to settle so the reagent could rise to the surface. Tubes were checked for the reddening of the alcohol layer within a few minutes, which indicated indole production [46].

#### 4.6.2. Phosphate Solubilization Activity of Endophytic Bacteria

Based on Pikovskaya [47] methods further created by Gupta et al. [48] which are widely applied as an evaluation tool [49], we assessed the potential of the isolates to solubilize phosphate on Pikovskaya’s agar medium (PVK). The PVK solid medium contains blue bromophenol, which produces halos around the colonies due to organic acid production. After inoculating fresh bacteria suspension (5 × 10^8^ CFU ml^−1^) on PVK agar media, the solubilization halo diameter around the colonies was estimated to determine bacteria solubilization ability. The solubilization index (SI) was calculated after 2, 7, and 10 days of incubation at 28 ± 2 °C using the following formula:$$SI=\frac{CD+HD}{CD}$$
where CD is the colony diameter, and HD is the halo zone diameter. All experiments were conducted with three repetitions [50].

#### 4.6.3. Nitrogen Fixation Activity of Endophytic Bacteria

The nitrogen fixation test was conducted by utilizing Ashby’s N-free medium (NFM), composed of 10 g of mannitol, 0.2 g of KH_2_PO_4_, 0.2 g of MgSO_4_·_7_H_2_O, 0.2 g of NaCl, 0.1 g of CaSO_4_·_2_H_2_O, 5 g of CaCO_3_, and adjusted to pH 7.0–7.5 with 1.8 g of agar in 1 L of distilled water [51]. Bacteria strains displaying normal growth on NFM plates were identified as potential nitrogen fixers [52]. *Escherichia coli* and Rhizobium bacteria are considered as the negative and positive control, respectively.

#### 4.6.4. Impact of Endophytic Bacteria on Pea Seed Germination

##### Seed Inoculation with Endophytic Bacteria

The seeds were surface sterilized in 70% (*v*/*v*) ethanol for 1 min followed by 10 min putting in 1% (*v*/*v*) sodium hypochlorite solution (Naclo-74.44 g/mol) and then washed six–seven times with sterile distilled water.

The suspensions of bacteria were prepared in LB medium and incubated overnight at 28 °C. After centrifugation at 5000× *g* for 5 min, the pellets were rinsed 2–3 times with sterile distilled water to remove any residual culture medium. Diluted with sterile 0.03 M MgSO_4_ and adjusted the concentration to 10^8^ cfu mL^–1^ by spectrophotometer at 600 nm. Pea seeds were soaked in a suspension of bacteria containing sterile 0.03 M MgSO_4_ for 4 h. The seeds were air-dried under laminar airflow conditions for 2 h.

##### Determination of Seed Germination and Plant Growth

The seed germination test was conducted by placing four inoculated seeds in transparent test containers. The transparent test containers were filled with wet sand and covered with black filter paper according to the Phytotoxkit protocol [53]. The seeds were put 0.5–1.0 cm from the top of the transparent containers. Four containers were used for each treatment, and they were placed in a growth chamber (16 h light/ 8 h darkness) at 23 °C. The percentage of germination was recorded after 48 h. Root and shoot length were measured after the ten days.

### 4.7. Statistical Analysis

In this study, statistical analyses were conducted through analysis of variance (ANOVA) using R studio statistical software (v 4.2.3). A post hoc test (Tukey’s HSD) was performed to test the significance of the differences among the treatments (*p* < 0.05).

## 5. Conclusions

This study underscored the significant pathogen inhibition and plant growth enhancement potential of endophytic bacteria isolated from different *Artemisia* species. The diversity of endophytic bacteria, including strains of *Bacillus* and *Pseudomonas*, highlights their functional importance in promoting plant health. The growth inhibition activity against *Fusarium* c.f. *oxysporum* by strains like *B. thuringiensis*, *B. cereus*, *B. velezensis*, *B. amyloliquefaciens*, and *Pseudomonas fluorescens* underscores their role as effective biocontrol agents. Furthermore, the biochemical properties, including phosphate solubilization, nitrogen fixation, and indole production, particularly by strain AR11, demonstrate their potential as plant growth-promoting agents. AR11 emerged as a promising candidate for further investigation as a microbial biofertilizer, potentially offering an environmentally friendly alternative to chemical pesticides for crop protection.

## Figures and Tables

**Figure 1 plants-13-01684-f001:**
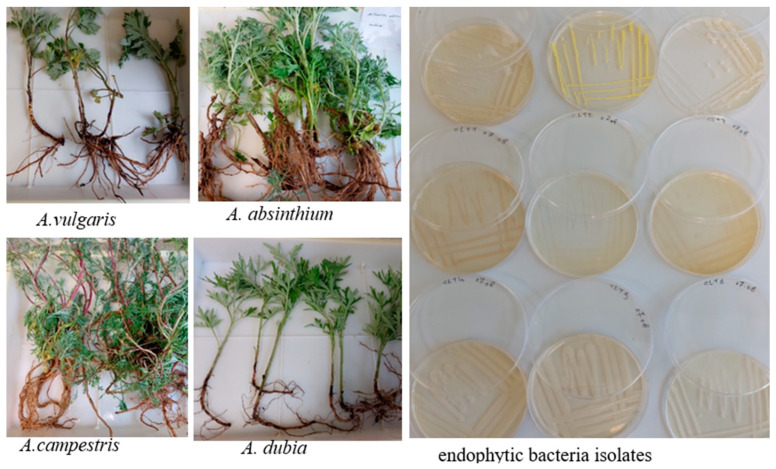
Four distinct species of *Artemisia* plants and some endophytic bacteria were isolated with different colony shapes, colors, and margins.

**Figure 2 plants-13-01684-f002:**
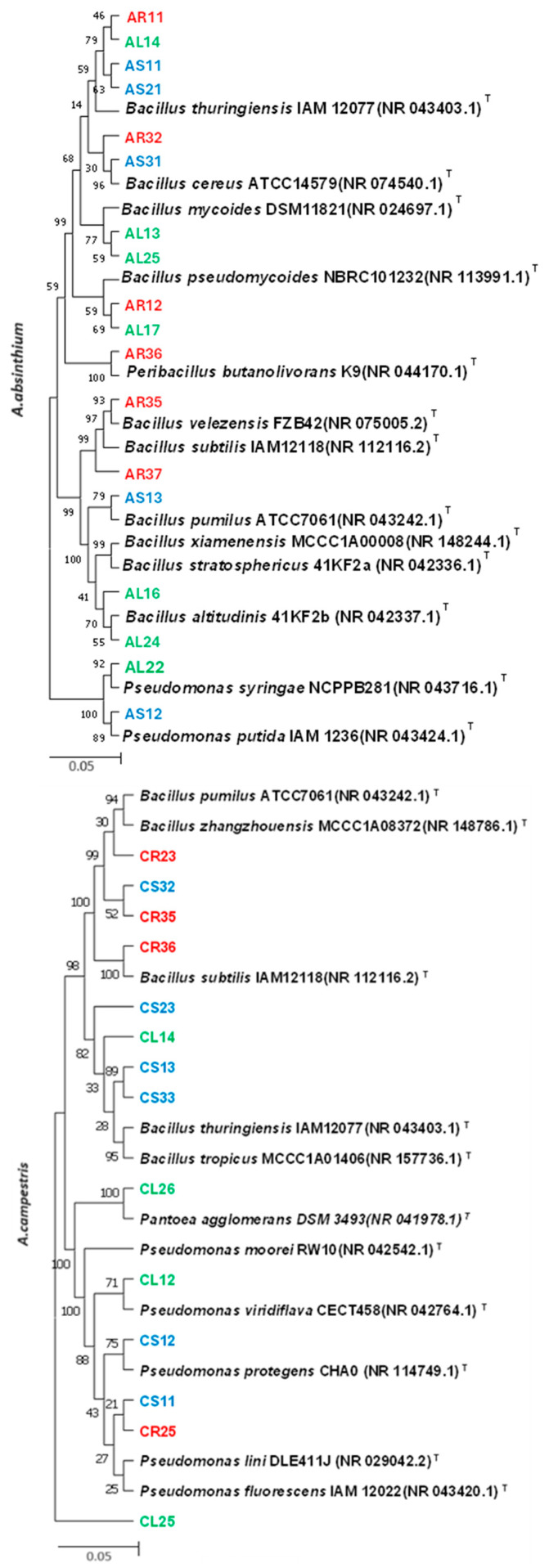
Phylogenetic tree reconstruction based on 16S rDNA sequences of endophytic bacteria isolated from *A. absinthium*, *A. campestris*, *A. dubia*, and *A. vulgar*. Neighbor joining was used to infer evolutionary history. The optimum tree is presented. The percentage of replicate trees with the relevant taxa clustered together in the bootstrap test (1000 repetitions) is displayed next to the branches. The sum of branch length for *A. absinthium* was 0.46909484, *A. campestris* 2.14063322, *A. dubia* 039531614, and *A. vulgaris* 2.39212283. Distinct strains, isolated from various plant parts such as roots, leaves, and stems, were differentiated by color, with strains from roots represented in red, leaves in green, and stems in blue.

**Figure 3 plants-13-01684-f003:**
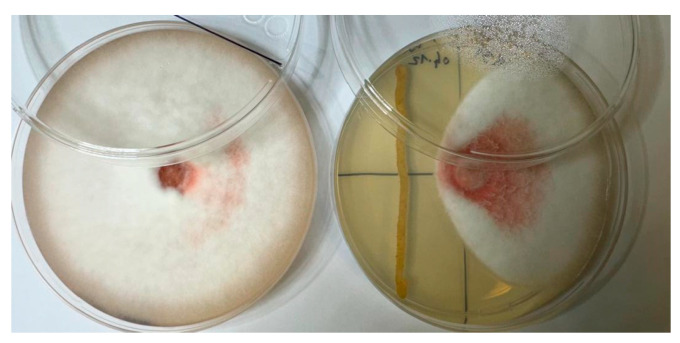
A dual culture method to determine the inhibitory effect of endophytic bacteria isolates on *Fusarium* spp. Inhibitory activity of AR11 against *F.* c.f. *oxysporum*—right, control plate—left.

**Figure 4 plants-13-01684-f004:**
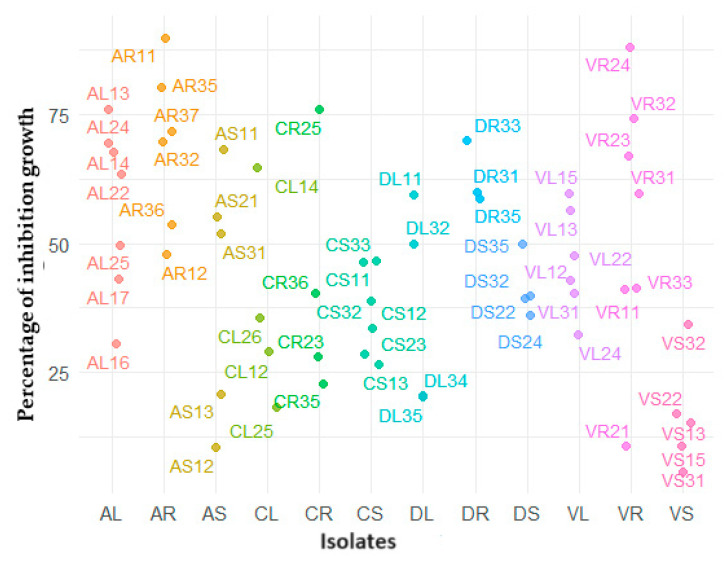
Percentage growth inhibition of endophytic bacteria on *F.* c.f. *oxysporum*. Significant differences were observed in the ability of endophytic bacteria against *F.* c.f. *oxysporum* (*p* < 0.001) on PDA media at 28 °C. Each data point represents the mean value of three replicates. The endophytic bacteria are grouped according to the plant parts they were isolated from as follows: AL (*A. absinthium*, leaf), AR (*A. absinthium*, root), AS (*A. absinthium*, stem), CL (*A. campestris*, leaf), CR (*A. campestris*, root), CS (*A. campestris*, stem), DL (*A. dubia*, leaf) DR (*A. dubia*, root) DS (*A. dubia*, stem), VL (*A. vulgaris*, leaf), VR (*A. vulgaris*, root), VS (*A. vulgaris*, stem).

**Figure 5 plants-13-01684-f005:**
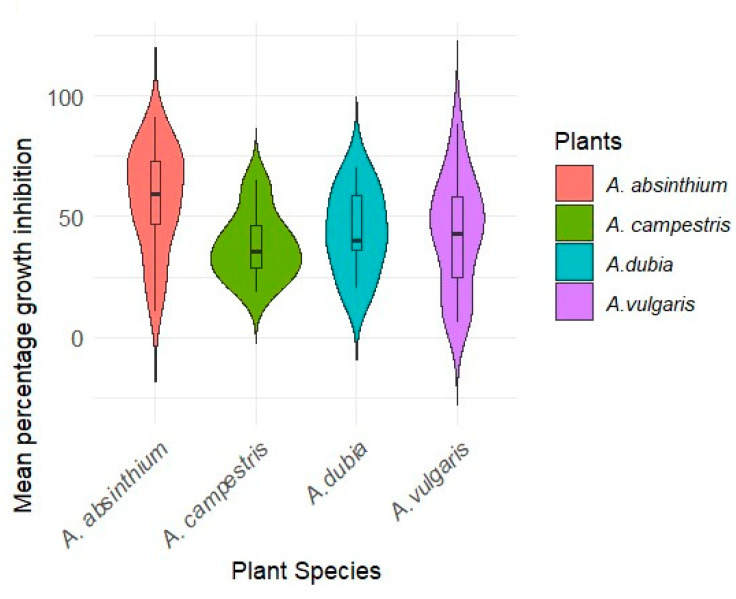
A comparison of the isolated endophytic bacteria to inhibit *Fusarium* c.f. *oxysporum* per plant. Data are based on mean percent inhibition across all isolates, the width of the violin plot illustrates the range of growth inhibition potential values within each *Artemisia* species.

**Figure 6 plants-13-01684-f006:**
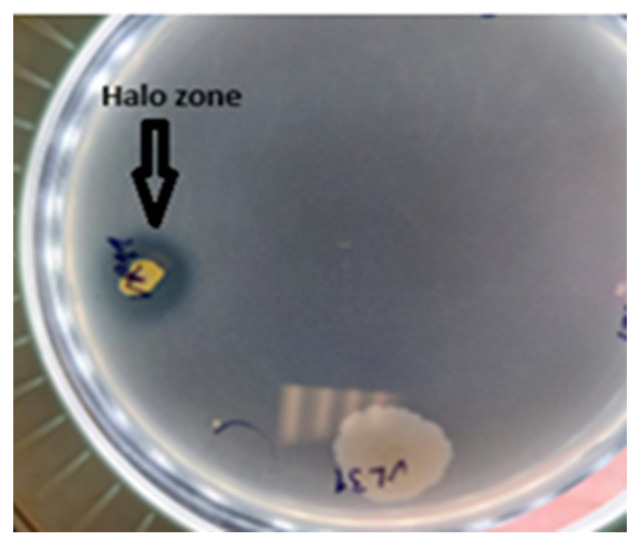
Formation halo zone by phosphate-solubilizing bacteria on Pikovskaya agar plate.

**Figure 7 plants-13-01684-f007:**
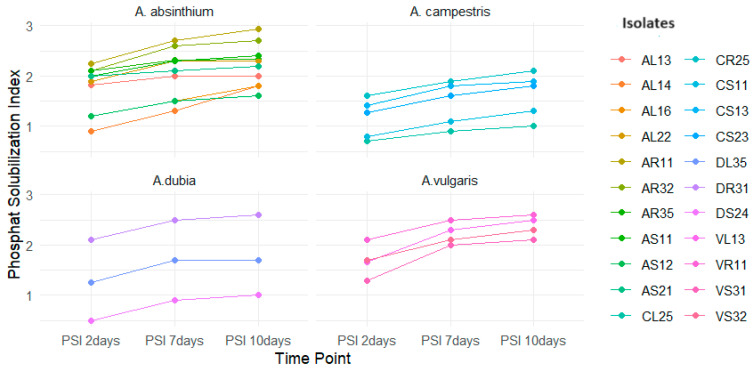
Phosphate solubilization index of isolates over the specified time points (2, 7, and 10 days). The solubilization index (SI) was computed by evaluating the ratio of the sum of colony diameter (CD) and halo zone diameter (HD) to the colony diameter (SI = (CD + HD)/CD).

**Figure 8 plants-13-01684-f008:**
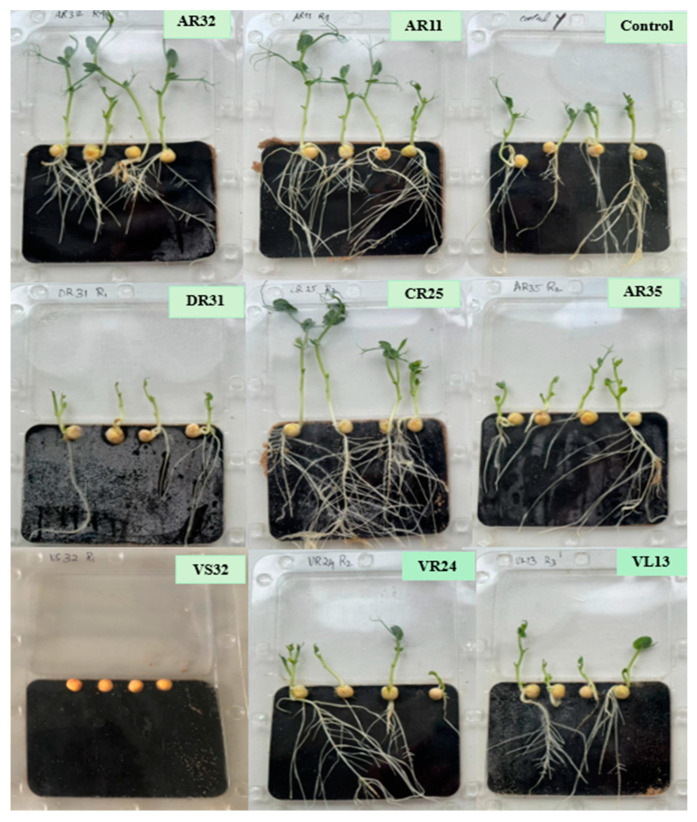
Effect of different endophytic bacteria strains on pea plant growth and root branching after 10 days.

**Figure 9 plants-13-01684-f009:**
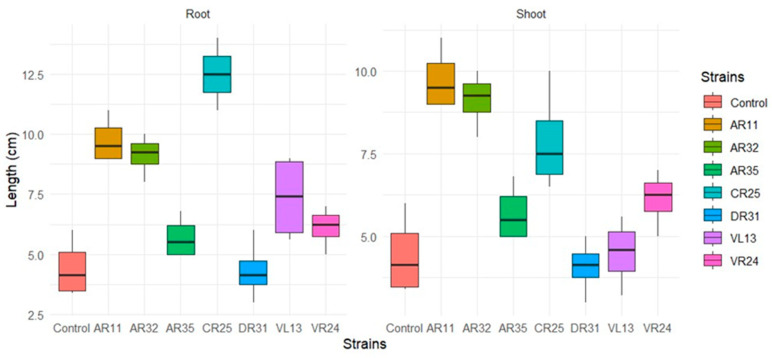
Effect of different bacteria strains on shoot and root length compared to control after 10 days.

**Figure 10 plants-13-01684-f010:**
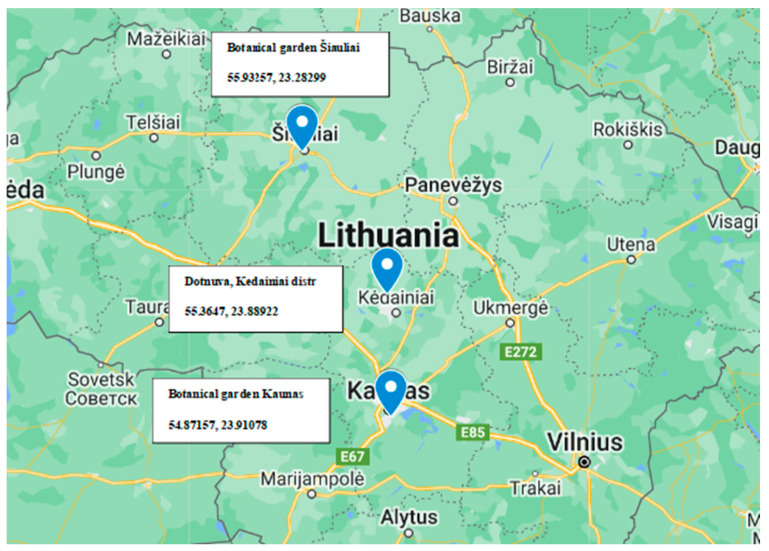
Geographical coordinates of *Artemisia* spp. used for strain isolation.

**Table 1 plants-13-01684-t001:** Characteristics of selected endophytic bacteria.

Isolate	Plant Species, Parts	Related Species	Similarity	Growth Inhibition	PSI	Indole Production Test	Nitrogen Fixation Test
AR11	*A. absinthium*, root	*Bacillus thuringiensis* IAM12077	100%	90.83%	2.93	+	+
AR32	*A. absinthium*, root	*Bacillus cereus* ATCC14579	99.57%	83.8%	2.7	−	+
AR35	*A. absinthium*, root	*Bacillus velezensis* FZB42	100%	80.8%	2.33	−	+
VR24	*A. vulgaris*, root	*Bacillus thuringiensis* IAM12077	99.91%	88%	-	−	+
VL13	*A. vulgaris*, leaf	*Bacillus cereus* ATCC14579	99.98%	56.36%	2.5	−	+
VS32	*A. vulgaris*, stem	*Bacillus cereus* ATCC14579	99.91	54.43%	2.3	−	+
CR25	*A. campestris*, root	*Pseudomonas fluorescens* IAM 12022	100%	75.86%	2.1	+	+
DR31	*A. dubia*, root	*Bacillus amyloliquefaciens* NBRC15535	100%	60%	2.6	−	+

(+/−) means positive and negative reaction to test.

## Data Availability

Data are contained within the article.

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
