# Peer review of "Exploring Endophytic Bacteria from Artemisia spp. and Beneficial Traits on Pea Plants"

_plants, 2024, doi:10.3390/plants13121684_

Round 1

Reviewer 1 Report (Previous Reviewer 2)

Comments and Suggestions for Authors

The work has been thoroughly revised. The analytical scope has been extended, which has led to interesting conclusions. I greatly appreciate the efforts of the authors who have proven that the cultured bacteria can be used as a basis for a biopreparation, as they have antifungal activity and can be used as a microbial biofertilizer.

I recommend accepting the work. I have found some minor errors, please remove them.

1. L18, 157, 171 and other: spp - use the spelling without italics.

2. L210, Figure 6: remove the “:” character.

3. L215, Figure 9 quoted before Figure 8 (L283). Swap the order of Figures 8 and 9.

4. L223. What do the abbreviations mean, e.g. AR32. Change the title to: Effect of different strains of endophytic bacteria on pea plant growth and root branching after 10 days.

5. L424-442. Was the efficiency of DNA isolation evaluated? Which method was used? What parameters did the DNA have? 

Author Response

Dear Reviewer

Thank you for your thorough review and positive feedback. We addressed and removed the minor errors you have noted. We appreciate your constructive comments and recommendation for acceptance.

  1. L18, 157, 171 and other: spp - use the spelling without italics. The edition has been done.
  2. L210, Figure 6: remove the “:” character. The edition has been done.
  3. L215, Figure 9 quoted before Figure 8 (L283). Swap the order of Figures 8 and 9. The edition has been done.
  4. L223. What do the abbreviations mean, e.g. AR32. Change the title to: Effect of different strains of endophytic bacteria on pea plant growth and root branching after 10 days. Abbreviations were removed, and the sentences were edited for more clarification. The title was changed according to the reviewer's comments. 
  5. L424-442. Was the efficiency of DNA isolation evaluated? Which method was used? What parameters did the DNA have? The details were added to clarify the methods used for checking the quality and quantity of DNA.

Reviewer 2 Report (Previous Reviewer 1)

Comments and Suggestions for Authors

The authors have made many changes to the manuscript, and it has improved however there is still issues with how the Fusarium isolates is being described, interpretation in the results, and the discussion would be greatly improved with subsection headers and some sentence on how the current results are similar to other results beyond just citation. When the authors cite other works and say they are similar. Please talk little about the other research, and what they found that was similar. Specific comments are within the manuscript.

As for the pathogenic Fusarium isolates. I believe the first submission the authors called it Fusarium oxysporum based on ITS1 and morphology but that is not possible due to the limitation of the ITS1 region. This version has the Fusarium labels in all but one place as Fusarium spp. suggesting many Fusarium species, not one. If the morphology of all the Fusarium isolates and DNA match favorably to Fusarium oxysporum, I would suggest calling it Fusarium c.f. oxysporum.

Comments on the Quality of English Language

The style of writing suggests some language barriers but mainly in a few places. In the discussion, subsection headers would help the reader understand the different section of the discussion.

Author Response

Dear Reviewer,

Thank you for your thorough review and valuable feedback. We have carefully considered your suggestions and made the following revisions to our manuscript:

Fusarium Identification:

  • Based on your comments, we have revised the terminology for the Fusarium isolates throughout the manuscript. We now refer to them as Fusarium c.f. oxysporum.

Discussion

  • We have significantly revised the discussion section to provide more detailed interpretations of our results. We have structured the discussion into distinct paragraphs that address different aspects of our findings.

Textual Revisions:

  • We have addressed all specific comments within the text.

Round 2

Reviewer 2 Report (Previous Reviewer 1)

Comments and Suggestions for Authors

The current version is well written and includes the previous suggested changes.

Comments on the Quality of English Language

NA

Author Response

Dear Reviewer,

Thank you for your review. In response to your suggestion for minor editing in English, some revisions have been made.

This manuscript is a resubmission of an earlier submission. The following is a list of the peer review reports and author responses from that submission.

Round 1

Reviewer 1 Report

Comments and Suggestions for Authors

Exploring Endophytic Bacteria from Artemisia spp.: Antagonistic Potential Against Pathogens and Contributions to plant growth promotion

The authors isolated endophytic bacteria from 4 species of Artemisia, and performed some antagonist studies.  The article is interesting, and they have completed a lot of work. At the core, it is scientifically sound but there are some issues that need to be addressed in this version of the manuscript.

1)      The results section should just state the results. There appears to be information that is more methods. For examples only, entire section should be reviewed:

Line 67: The isolates underwent purification through subculturing methods.

Lines 76-77: To preliminarily identify bacterial endophytes, we conducted a thorough analysis of their morphological, biochemical, and physiological traits.

The results should only discuss the results in a clear and concise manner.

2)      Sentences that are vague: Below is an example for what I mean. The results should use concreate terminology.

Line 99-102: “A notable outcome of our study was the identification of sixty-one bacterial strains exhibiting significant antagonistic efficacy against the fungal pathogens. Quantification of mycelium growth suppression revealed a spectrum of inhibitory effects, with isolates displaying a wide range of activities against both F. solani and F. oxysporum.”

1) What is quantification of mycelium growth?

2) A spectrum of inhibitory effects?

3) a wide range of activities?

Line 142-143: “Bacterial strains obtained from A. absinthium manifest a spectrum of inhibitory activities distinct from those isolated from A. campestris and A. vulgaris.”

4) manifest a spectrum?

5) distinct?

The results should be concrete statements that demonstrate these sentences. These sentences are more like a discussion. In in general, they do not impart concrete information.

3)      The authors have identified both Fusarium species and using the ITS2, partial LSU region.

This is insufficient for species level determination for Fusarium species. The authors can state the best match, but they should take care not to call them F. solani and F. oxysporum without further molecular analysis. Perhaps, if the morphology matches, the authors should refer to them as: F. cf. solani and F. cf. oxysporum, this at least implies some uncertainty which does exist at this point. However, the authors need to state that both the molecular and morphological characteristics match those species of imply cf.

1)See: https://web.archive.org/web/20150402154620/http://cdn.palass.org/publications/palaeontology/volume_31/pdf/vol31_part1_pp223-227.pdf

2)https://link.springer.com/article/10.1007/s00253-011-3209-3

“The sequences most commonly used to distinguish Fusarium spp. are portions of the genomic sequences encoding the translocation elongation factor 1-α (TEF) (Wulff et al. 2010), β-tubulin (tub2) (O’Donnell et al. 1998a), calmodulin (O’Donnell et al. 2000), internally transcribed spacer regions in the ribosomal repeat region (ITS1 and ITS2) (Waalwijk et al. 1996; O’Donnell and Cigelnik 1997), and the intergenic spacer region (IGS) (Yli-Mattila and Gagkaeva 2010). Not all sequences work equally well for all species, with tef1 gene being the most widely accepted across the genus. The ITS regions do not work well within the Liseola section and also for many other Fusarium species, such as Fusarium avenaceum, Fusarium arthrosproioides/Fusarium tricinctum, F. sporotrichioides/F. langsethiae, and the lineages of F. graminearum species complex, and its near relatives (O’Donnell et al. 2000; Yli-Mattila et al. 2002). Also, β-tub2 was reported to not work well within the Fusarium solani species complex (Sampietro et al. 2010). Genes encoding other proteins (e.g., histone H3, calmodulin, and Tri101) and mating type among others have also been used to distinguish species in different portions of the genus (Mule et al. 2005). In practice, it is difficult to distinguish Fusarium species from its close relatives, and accurate identification based on traditional methods is very difficult due to their genetic variation and high morphological similarity.”

4)      All Figures should have the titles removed on top, and have solid Figure titles. There is no need to state that they are scatter plots or bar charts.

5)      The methods are too vague in places. The primers need citations, and the amounts of inoculation need to be discussed particularly with the assays as the amount of inoculation can alter the results.

Overall: I do like this experiment, but the manuscript needs to be clear and concise. Using a lot hype words (see below) obscures the information and only makes it harder to read. For example, was the biplot analysis really “effectively deployed” and if so, what does that really mean. I think there is a lot of potential, just have clear concise sentences, and limit the use of distracting word.

Lines 173-175: “Biplot analysis was effectively deployed to visualize the intricate interrelationships and patterns inherent in various plant species, plant parts, and their corresponding antagonistic effects on the inhibition growth percentages of F. oxysporum and F. solani.

There is a lot of unnecessary words and if you read it, there is little information to the reader.

Attached are some additional minor issues to address.

Comments on the Quality of English Language

There are some issues in a few sections, but overall ok. Please see the attachment.

Author Response

We would like to thank the reviewers for taking the time to review our manuscript and providing valuable feedback on our work. We have addressed the issues raised in the attached response to the comments document.

Reviewer 2 Report

Comments and Suggestions for Authors

Review of the manuscript entitled "Exploring Endophytic Bacteria from Artemisia spp.: Antagonistic Potential Against Pathogens and Contributions to plant growth promotion” by Hadian et al.

The assumptions made by the authors are interesting and valuable. However, the paper lacks refinement and the results should only be considered preliminary. In my opinion, the publication is premature. To improve the value of their work, the authors should perform species identification of the bacteria based on multiple genes and conduct a biochemical profile analysis of the strains. Additionally, they should investigate what caused the activity of the bacterial isolates against fungi, as it is not relevant to know where they were isolated from or the plant species they were isolated from. Without these analyses, the results are of little value. This is my assessment of the work. I suggest that the paper should be rejected for publication in its current form.

General comments:

1) The title of the thesis is not appropriate, it should show the most important result of the work to encourage the potential reader to read the content of the thesis.

(2) In the paper's abstract, please state the proper scope of the research work. There is information on the identification and characterisation of endophytic bacteria from four Artemisia plant species. What methods were used in the study?

(3) Conclude that trains belonging to the Bacillus genus exhibited the greatest antagonistic activity. Explain which species the strains belonged to (AR11 and VR24). Their characteristics should be shown. The fact that they were isolated from roots is interesting, but it is not a conclusion.

(4) The description of the results should follow the chronology of the activity:

- Isolate and identify the diversity of endophytic bacteria from different plant parts of four species of Artemisia spp.

- molecular and biochemical identification of isolated endophytic bacteria;

- antagonistic activity of the isolated bacteria against F.oxysporum, F.solani

- assess the phosphate solubilization ability and indole production of the isolated strains

The methodology should be shown analogously.

(5) The lack of species identification of the bacteria makes the results preliminary and not worthy of publication.

(6) The authors did not use the method of analysis of variance in their study. At least I did not find such results.

(7) What are the specific conclusions of the analyses performed by the Authors. Representatives of which species had the strongest antagonistic properties towards Fusarium? How do the authors explain this phenomenon? What are the biochemical characteristics of these bacteria?

Specific comments:

1. L91-98 This is not a description of the antagonism test.

2. 2.2. Antagonistic activity of endophytic isolate on pea pathogen.

The results of both tests should be shown.

3. L105-106 I do not see the results of the statistical analysis.

4. Fig. 4 adds nothing to the analysis. The results are just a curiosity, not valuable for analysis. The authors set the goal of assessing the diversity of endophytic bacteria. I do not see the diversity indices calculated.

5 Provide geographical coordinates identifying the study area (GPS).

6. L399-404. give pea variety, date of plant collection,. place of collection. On which sample was the study conducted? Were only two fungal isolates isolated? Give isolate codes and place of culture storage (collection).

7. L423. What are conidiospores?

8. L449. What sequencing method was used?

9. L462. Give the names of the isolates and the species. Was the pathogenicity of the strains isolated from peas tested? If so what was it?

10. l459-470. In how many replicates was the study carried out? How many experimental series were there? Photographic documentation should be attached to the paper.

11. L479. The title is not appropriate.

12. Fig.2, Fig.3, Are not legible. No statistics. Bacterial species should be identified rather than strain numbers. Are there species that inhibited the growth of both fungi?

13. Fig. 10, What does Sample ID mean? What do these results contribute to the analysis without specifying species?

14. The discussion of the results is weak. What does the occurrence of bacterial species depend on? No discussion in relation to bacterial species identification. No discussion about antagonism tests.

Author Response

Dear Reviewer,

Thank you for your feedback. We have carefully considered your comments and have addressed them in the attached response document.

Round 2

Reviewer 1 Report

Comments and Suggestions for Authors

The authors have made many corrections, but the underlying issue of clarity is still problematic and not a matter of writing style. The main issue at hand is the manuscript has separate methods, results, and discussion sections (not a unified results and discussion section). Currently, the results section contains both methods, and interpretation which makes the article appear unorganized as written, particularly within the results section. Please avoid sentence that sound like methods or sentences that are interpretations. I have provided some examples in the comments in the PDF.

Also, there are at least two figures without Figure legends in this version, and all figures should be stand-alone (meaning its ok to repeat what each acronym means in multiple figures for clarity).

The methods also need to be tightened up and repeatable. This means, shaker speeds, microscopes used to evaluate morphology, etc. Please see the comments within the PDF.

Comments on the Quality of English Language

minor throughout, see comments in PDF

Author Response

Dear reviewer,

We appreciate your valuable feedback and guidance throughout the review process. We have carefully considered each of your comments and made the necessary revisions to improve the clarity and organization of the manuscript. All the editing has been done according to the review comments on the PDF file. We have removed sentences that sound like methods from the results section. Regarding the pictures without legends, they have already been removed from the manuscript according to the second reviewer's suggestion. Now, for further clarification, those figures have been completely deleted. Additionally, we have included additional details such as shaker speeds, microscopes used for morphology evaluation, and other relevant information.

Thank you for your time and consideration.

Reviewer 2 Report

Comments and Suggestions for Authors

I appreciate the authors' efforts to improve the quality of the text. However, without a determination of the species affiliation of the antagonistic bacteria and an explanation of the mechanism of the antagonistic effect on Fusarium fungi, the results presented are preliminary. The plant species, the place of isolation and the plant organ do not determine the antagonistic properties of the bacteria. The study authors' reference to the fact that the isolated bacteria are ecologically balanced is interesting. Still, in this case the bacterial populations isolated from the same plant species in different locations need to be compared and then their functions in the microbiome weighed up. Again, this is a preliminary study without being able to identify the species of each isolate. To summarise, I stand by my assessment. In its current form, the study is not suitable for publication in a high-ranking Plants journal.

Author Response

Thank you for your feedback. We have carefully considered your comments and made all the necessary changes accordingly.

Our study provides valuable insights into the potential role of endophytic bacteria in sustainable agriculture by isolating, identifying, and characterizing endophytic bacteria from multiple Artemisia plant species. We present novel findings on the isolation and characterization of endophytic bacteria from four different Artemisia species (A. absinthium, A. campestris, A. dubia, and A. vulgaris), for which there are no previous reports regarding endophytic bacteria association on these plant species. By focusing on endophytic bacteria from distinct plant parts of Artemisia species across three diverse locations, we aimed to clarify the diversity and potential applications of these microorganisms in sustainable agriculture.

We focused on 62 isolates with the potential to inhibit the growth of pathogenic fungi out of eighty-four endophytic bacterial isolates. Despite the diversity in plant species, plant parts, and sampling locations, our results revealed a consistent trend: beneficial traits predominantly concentrated in bacteria belonging to the genera Bacillus and Pseudomonas. This insight not only highlights the importance of these bacterial genera in endophytic communities but also provides valuable guidance for future research aimed at utilizing their potential for agricultural applications. This innovative work offers clarity on which Artemisia species and associated endophytic bacteria desert further investigation, paving the way for targeted studies to explore their potential as sustainable agricultural solutions.

Furthermore, we acknowledge the need for further molecular analysis, including whole-genome sequencing of selected isolates, to better understand their functional potential. Our future plans include conducting whole-genome sequencing and further molecular analysis on a limited number of isolates that have shown effectiveness in our current research. However, it is not feasible to conduct such analyses on all 62 isolates.

Overall, our study underscores the multifaceted benefits of endophytic bacteria associated with Artemisia plants in sustainable agriculture and has significant implications for agricultural research aimed at developing microbe-based biofertilizers or biopesticides. It demonstrates that endophytic bacteria associated with Artemisia have the potential to be further investigated for the production of biofertilizers or biopesticides. Thank you for your consideration.